# Reference Values to Assess Hemodilution and Warn of Potential False-Negative Minimal Residual Disease Results in Myeloma

**DOI:** 10.3390/cancers13194924

**Published:** 2021-09-30

**Authors:** Noemí Puig, Juan Flores-Montero, Leire Burgos, María-Teresa Cedena, Lourdes Cordón, José-Juan Pérez, Luzalba Sanoja-Flores, Irene Manrique, Paula Rodríguez-Otero, Laura Rosiñol, Joaquín Martínez-López, María-Victoria Mateos, Juan-José Lahuerta, Joan Bladé, Jesús F. San Miguel, Alberto Orfao, Bruno Paiva

**Affiliations:** 1Hematology Department, Hospital Universitario de Salamanca (HUSAL), IBSAL, IBMCC (USAL-CSIC), CIBERONC (CB16/12/00233), 37007 Salamanca, Spain; jpmoran@saludcastillayleon.es (J.-J.P.); mvmateos@usal.es (M.-V.M.); 2Institute of Biomedical Research of Salamanca (IBSAL), Centro de Investigación Biomédica en Red Cáncer CIBERONC (CB16/12/00400), Cancer Research Center (IBMCC, USAL-CSIC), Cytometry Service (NUCLEUS) and Department of Medicine, University of Salamanca, 37007 Salamanca, Spain; jflores@usal.es (J.F.-M.); orfao@usal.es (A.O.); 3Clínica Universidad de Navarra, Centro de Investigación Aplicada (CIMA), Instituto de Investigación Sanitaria de Navarra (IDISNA), CIBERONC (CB16/12/00369), 31008 Pamplona, Spain; lbrodriguez@unav.es (L.B.); imsaenzdete@unav.es (I.M.); paurodriguez@unav.es (P.R.-O.); sanmiguel@unav.es (J.F.S.M.); bpaiva@unav.es (B.P.); 4Hospital Universitario 12 de Octubre, CIBERONC (CB16/12/00369), 28041 Madrid, Spain; mariateresa.cedena@salud.madrid.org (M.-T.C.); jmarti01@med.ucm.es (J.M.-L.); jjLAHUERTA@telefonica.net (J.-J.L.); 5Grupo de Investigación en Hematología, Instituto de Investigación Sanitaria La Fe (IIS La Fe), CIBERONC (CB16/12/00284), 46026 Valencia, Spain; lourdes_cordon@iislafe.es; 6Institute of Biomedicine of Seville, Department of Hematology, University Hospital Virgen del Rocío of the Consejo Superior de Investigaciones Científicas (CSIC), University of Seville, Seville, Spain & CIBERONC (CB16/12/00480), Instituto Carlos III, 28029 Madrid, Spain; lucysanoja@usal.es; 7Hematology Department, IDIBAPS, Hospital Clinic, 08036 Barcelona, Spain; LROSINOL@clinic.cat (L.R.); JBLADE@clinic.cat (J.B.)

**Keywords:** multiple myeloma, minimal residual disease, hemodilution

## Abstract

**Simple Summary:**

Although the majority of patients with myeloma who achieve undetectable minimal residual disease show prolonged survival, some of them relapse shortly afterwards. False-negative results due to hemodiluted bone marrow samples could explain this inconsistency, but there is no guidance on how to evaluate them. We analyzed three cell populations normally absent in peripheral blood in 1404 aspirates obtained in numerous disease settings and in 85 healthy adults. Pairwise comparisons according to age and treatment showed significant variability, thus suggesting that hemodilution should be preferably evaluated with references obtained after receiving identical regimens. Leveraging the minimal residual disease results from 118 patients, we showed that a comparison with age-matched healthy adults could also inform on potential hemodilution. Our study supports the routine assessment of bone marrow cellularity to evaluate hemodilution, using as reference values either treatment-specific or from healthy adults if the former are unavailable.

**Abstract:**

Background: Whereas, in most patients with multiple myeloma (MM), achieving undetectable MRD anticipates a favorable outcome, some others relapse shortly afterwards. Although one obvious explanation for this inconsistency is the use of nonrepresentative marrow samples due to hemodilution, there is no guidance on how to evaluate this issue. Methods: Since B-cell precursors, mast cells and nucleated red blood cells are normally absent in peripheral blood, we analyzed them in 1404 bone marrow (BM) aspirates obtained in numerous disease settings and in 85 healthy adults (HA). Results: First, we confirmed the systematic detection of the three populations in HA, as well as the nonreduced numbers with aging. Pairwise comparisons between HA and MM patients grouped according to age and treatment showed significant variability, suggesting that hemodilution should be preferably evaluated with references obtained from patients treated with identical regimens. Leveraging the MRD results from 118 patients, we showed that a comparison with HA of similar age could also inform on potential hemodilution. Conclusions: Our study supports the routine assessment of BM cellularity to evaluate hemodilution, since reduced BM-specific cell types as compared to reference values (either treatment-specific or from HA if the former are unavailable) could indicate hemodilution and a false-negative MRD result.

## 1. Introduction

Measurable residual disease (MRD) is routinely evaluated in multiple myeloma (MM) clinical trials [1,2,3,4,5,6]. The importance of undetectable MRD to achieve long-term survival was recently confirmed in a large meta-analysis [7], and MRD response rates have been the endpoint in numerous clinical trials [1,2,3,8,9,10]. However, the role of MRD to refine patients’ management in routine practice remains controversial.

The uncertainty about the meaning of undetectable MRD at the individual patient level holds back the use of MRD to guide treatment decisions in MM; whereas, in most cases, it predicts a favorable outcome, some patients continue to relapse after achieving MRD negativity in “real-world” studies, as well as in clinical trials [11,12]. Such a discordance between laboratory and clinical findings could possibly be attributed to the disease biology (e.g., a high proliferation of residual cells), as well as to low-sensitive assays, extramedullary disease and to nonrepresentative bone marrow (BM) aspirates either due to patchy tumor infiltration or hemodilution [2].

The advent of next-generation sequencing (NGS) and flow (NGF) cytometry enabled sensitivity levels of 10^−5^ and beyond in clinical trials [13,14] and routine practice [11,12,15]. Positron emission tomography-computed tomography has a complementary role in BM MRD assessments in patients with MM and may identify false-negative results due to intra- and extramedullary diseases not depicted by NGS or NGF [16,17]. However, until now, there has been no guidance on how to evaluate the extent of hemodilution in BM aspirates used for MRD assessment in MM. Thus, we aimed to resolve this limitation by providing reference values of key BM cell types for ready applications in numerous disease settings.

## 2. Materials and Methods

### 2.1. Patients and Treatment

This study included 666 patients diagnosed with MM and 85 healthy adults (HA) (median age 45; range, 19–96 years). Patients were enrolled in the PETHEMA/GEM2012MENOS65 (*n* = 408), Ref. [18] PETHEMA/GEMCLARIDEX (*n* = 63) [19] and PETHEMA/GEM2017FIT (*n* = 22) clinical trials for the initial treatment of transplant-eligible and ineligible MM, as well as in the PETHEMA/GEM2014MAIN (*n* = 153) maintenance study. An additional cohort of 20 Spanish MM patients treated with anti-BCMA CAR T cells was also included. Each clinical trial was approved by the Spanish Agency of Drugs and Sanitary Products. These were registered at www.clinicaltrials.gov (accessed on 11 August 2021) (NCT01916252, NCT02575144, NCT03742297 and NCT02406144; accessed on 1 August 2021) and were conducted as per the ethical principles of the Declaration of Helsinki. Signed informed consent forms were required prior to patient enrollment in the clinical trials. This study was approved by the Ethics Committee of the Clinica Universidad de Navarra (164/2015).

The PETHEMA/GEM2012MENOS65 was an open-label phase 3 study that included 458 transplant-eligible patients receiving six induction cycles of bortezomib, lenalidomide and dexamethasone (VRD) autologous stem cell transplant (ASCT) after high-dose therapy (HDT) with either Bu-Mel or with Mel-200 and two consolidation cycles of VRD [18]. Afterwards, the patients were offered to be enrolled in the PETHEMA/GEM2014MAIN clinical trial that randomized their maintenance with RD or RD plus ixazomib for two years [14]. The PETHEMA/GEMCLARIDEX clinical trial was an open-label phase 3 study that included 286 transplant-ineligible patients randomly assigned in a 1:1 ratio to receive Rd or Rd plus clarithromycin [19]. All patients were treated until disease progression or unacceptable toxicity. The PETHEMA/GEM2017FIT clinical trial is a phase 3 study in which patients are randomly assigned in a 1:1:1 ratio to receive 18 induction cycles with bortezomib, melphalan and prednisone (VMP), followed by Rd vs. carfilzomib, lenalidomide and dexamethasone (KRd) with/without daratumumab (Dara-KRd), followed by consolidation and maintenance with lenalidomide and daratumumab. Twenty relapsed/refractory patients with MM treated at the Clínica Universidad de Navarra and the Hospital Universitario de Salamanca with anti-BCMA CAR T-cell therapy were also included in the study.

### 2.2. Sample Collection

A total of 1404 BM aspirates were analyzed: 85 from HA; 63 after induction with lenalidomide and dexamethasone (Rd) plus/minus clarithromycin (PETHEMA/GEMCLARIDEX); 300 after induction with VRD; 364 at day 100 after autologous transplant; 350 after two VRD consolidation courses (PETHEMA/GEM2012MENOS65); 34 after induction with carfilzomib, lenalidomide, dexamethasone and daratumumab (Dara-KRd) (PETHEMA/GEM2017FIT); 232 yearly during maintenance with Rd plus/minus ixazomib (PETHEMA/GEM2014MAIN) and 61 after CAR-T-cell infusion. All samples were analyzed using NGF cytometry following the EuroFlow standard operation procedures [20].

The impact of potential hemodilution of the BM aspirates was further evaluated in 118 transplant-ineligible MM patients enrolled in the PETHEMA/GEM2010MAS65 clinical trial (NCT01237249) with available MRD results and data on BM cellular composition at the end of treatment [21,22].

### 2.3. Assessment of MRD and Other BM Cell Types

MRD was predefined to be assessed using the EuroFlow NGF method [20,23] after induction, at day 100 after HDT/ASCT, after consolidation and annually during maintenance in transplant-eligible patients (PETHEMA/GEM2012MENOS65 and PETHEMA/GEM2014MAIN). In transplant-ineligible patients, MRD was analyzed after 18 induction cycles (PETHEMA/GEMCLARIDEX, PETHEMA/GEM2017FIT and PETHEMA/GEM2010MAS65 clinical trials). In patients treated with CAR T cells, MRD was predefined to be assessed at months 1, 3, 6 and 12 after infusion.

The first 8-color antibody combination (CD138-BV421, CD27-BV510, CD38-FITC, CD56-PE, CD45-PerCPCy5.5, CD19-PECy7, CD117-APC and CD81-APCH7) was used for the enumeration of mast cells (CD117bright and CD45dim); nucleated red blood cells (CD45–, CD38–, CD117–/+ and SSClo) and B-cell precursors (CD19+, CD45dim, CD38bright, CD81bright and CD27–) [20,24]. Data acquisition was performed in a FACSCanto II flow cytometer (BD, San Jose, CA, USA) using FACSDiva 6.1 software (BD). Data analysis was performed by experienced operators using Infinicyt software (Cytognos SL, Salamanca, Spain).

### 2.4. Statistical Analyses

The Mann–Whitney *U* and Χ^2^ tests were used to estimate the statistical significance of the differences observed between groups, and correlation studies were performed using the Pearson’s test. Survival curves were plotted by the Kaplan–Meier method and compared by the two-sided log-rank test. Progression-free survival was defined as the time from MRD assessment to disease progression or death from any cause and overall survival as the time from MRD assessment to death from any cause. Statistical analyses were conducted using Stata (version 15.0; StataCorp LP, TX, USA) and SPSS (version 20.0; IBM, Chicago, IL, USA).

## 3. Results and Discussion

The analysis of BM aspirates using EuroFlow NGF cytometry for MRD assessment in MM enables the systematic identification of ≥16 cell types [25]. We deliberately focused on B-cell precursors, mast cells and nucleated red blood cells (NRBC), since these are normally absent in peripheral blood and could therefore be useful in evaluating hemodilution (Appendix A). We first investigated a possible correlation between the ages and the percentages of these cell types in HA and only found a modest increment of mast cells with aging (Figure 1). Accordingly, there were no differences in the median percentages of the B-cell precursors and NRBC when comparing HA younger and older than 65 years, while the frequency of mast cells significantly increased in the latter (Table 1 and Figure 2). These findings further supported the evaluation of these cell types to assess hemodilution due to nonreduced numbers with aging and systematic detection in BM aspirates from HA.

Numerous drugs used in MM have immunomodulatory effects [26,27,28,29]. Thus, it could be hypothesized that reference values of B-cell precursors, mast cells and NRBC in HA may not represent those found in patients’ BM, particularly during treatment. However, the impact of therapy in the frequency of these cell types has never been investigated. Pairwise comparisons between HA and MM patients respectively grouped according to age and treatment uncovered that, among 36 possible comparisons per cell type, B-cell precursors displayed the highest variability during treatment (significant differences being found in 29 out of 36 comparisons, 80.5%), followed by NRBC (20/36, 55%) and mast cells (19/36, 53%) (Figure 2). B-cell precursors were notoriously decreased in BM aspirates from elderly MM patients receiving induction with Rd and Dara-KRd but not in transplant-eligible cases treated with VRD (Table 1). As expected, hematopoietic regeneration after autologous transplant resulted in a significant expansion of B-cell precursors that persisted after the consolidation, though not during maintenance therapy (Table 1). Mast cells were significantly increased during maintenance, whereas the NRBC were generally increased in BM aspirates of MM patients when compared to HA, except in those treated with anti-BCMA CAR T cells (Table 1 and Figure 2). These results suggested that, whenever possible, hemodilution should be evaluated considering reference values obtained from MM patients treated with a similar regimen.

We also investigated the potential utility of reference values from HA in cases where treatment-specific data are unavailable. To this end, we leveraged the MRD results and data on BM cellular composition 18 from 118 transplant-ineligible MM patients enrolled in the PETHEMA/GEM2010MAS65 clinical trial (Table 2) [22]. Namely, we stratified MRD-negative patients at the end of treatment according to the presence of each of the three BM cell types below or above the respective median value in HA older than 65 (Table 1). Only one MRD-negative patient displayed <2.5% NRBC, and further analyses based on this cell type were therefore unfeasible. Patients with undetectable MRD and <0.005% vs. ≥0.005% mast cells had a median progression-free survival (PFS) of 35 vs. 52 months, though the differences were not statistically significant (data not shown). Patients with undetectable MRD and <0.31% B-cell precursors displayed significantly inferior PFS vs. MRD-negative cases with ≥0.31% B-cell precursors (median of 28 vs. 55 months vs. not reached, *p* = 0.033; Figure 3A), as well as a trend towards a reduced overall survival (Figure 3B). Thus, while this study should foster the community to build upon the current 1404-sample large dataset by adding reference values found in other treatment scenarios, an immediate comparison with HA of similar ages could inform us of the potential BM hemodilution. That notwithstanding, there were no significant differences in the survival outcomes between MRD-negative patients after consolidation in the PETHEMA/GEM2012MENOS65 clinical trial, stratified based on percentages below or above the respective median value of B-cell precursors, mast cells and NRBC found in the same population (data not shown). Therefore, the utility of the reference values specific to certain regimens provided here should be confirmed by other groups.

In contrast to that found in HA where B-cell precursors, mast cells and NRBC were systematically detected in BM aspirates, these cell types were absent in significantly variable frequencies of MM patients (Table 3). B-cell precursors and mast cells were more commonly absent than NRBC. Of note, the simultaneous absence of all three cell types was significantly more frequent in MM patients treated with CAR T cells, which could be related to the hematological toxic effects of lymphodepleting chemotherapy. In such cases, severely hemodiluted samples would potentially explain the relatively high rate of patients with undetectable MRD by NGS, despite persistent M components after infusion with bb2121 CAR T cells (*n* = 7/17, 41%) [30]. Indeed, we observed patients in whom positive MRD testing was preceded by nonrepresentative BM aspirates that could lead to equivocal results unless B-cell precursors, mast cells and NRBC are also analyzed (Figure 4).

## 4. Conclusions

Our study supports the routine assessment of BM cellularity to assess hemodilution while evaluating MRD and provides reference values of B-cell precursors, nucleated red blood cells and mast cells in numerous treatment settings. These can be assessed in a reproducible manner with a reduced set of markers (i.e., CD19, CD27, CD38, CD45, CD81 and CD117). In patients with undetectable MRD, special attention should be paid to the percentage of BM-specific cell types, and in cases in whom these are reduced when compared to reference values (either treatment-specific or from HA if the former are unavailable), the report should indicate a possible risk of hemodilution and of a false-negative MRD result. However, a reduced percentage of an individual cell type may also reflect impaired hematopoiesis due to other factors (e.g., treatment or an altered tumor microenvironment). Therefore, a greater specificity in defining hemodiluted samples is warranted and could be potentially achieved by considering those where the complete or near-absence of all cell types is simultaneously observed. It should also be noted that, in patients with suspected hemodilution, an MRD-negative test could be either real or false. Thus, the possibility of performing mass spectrometry before repeating a BM aspirate should be considered, and the need of a new sample would be warranted only in cases where the M-protein remains undetectable. Taken together, a greater confidence in the robustness of MRD testing is a prerequisite for its progressive implementation in routine practice, and this study provides additional tools and a path forward to increase the level of confidence in MRD-negative results.

## Figures and Tables

**Figure 1 cancers-13-04924-f001:**
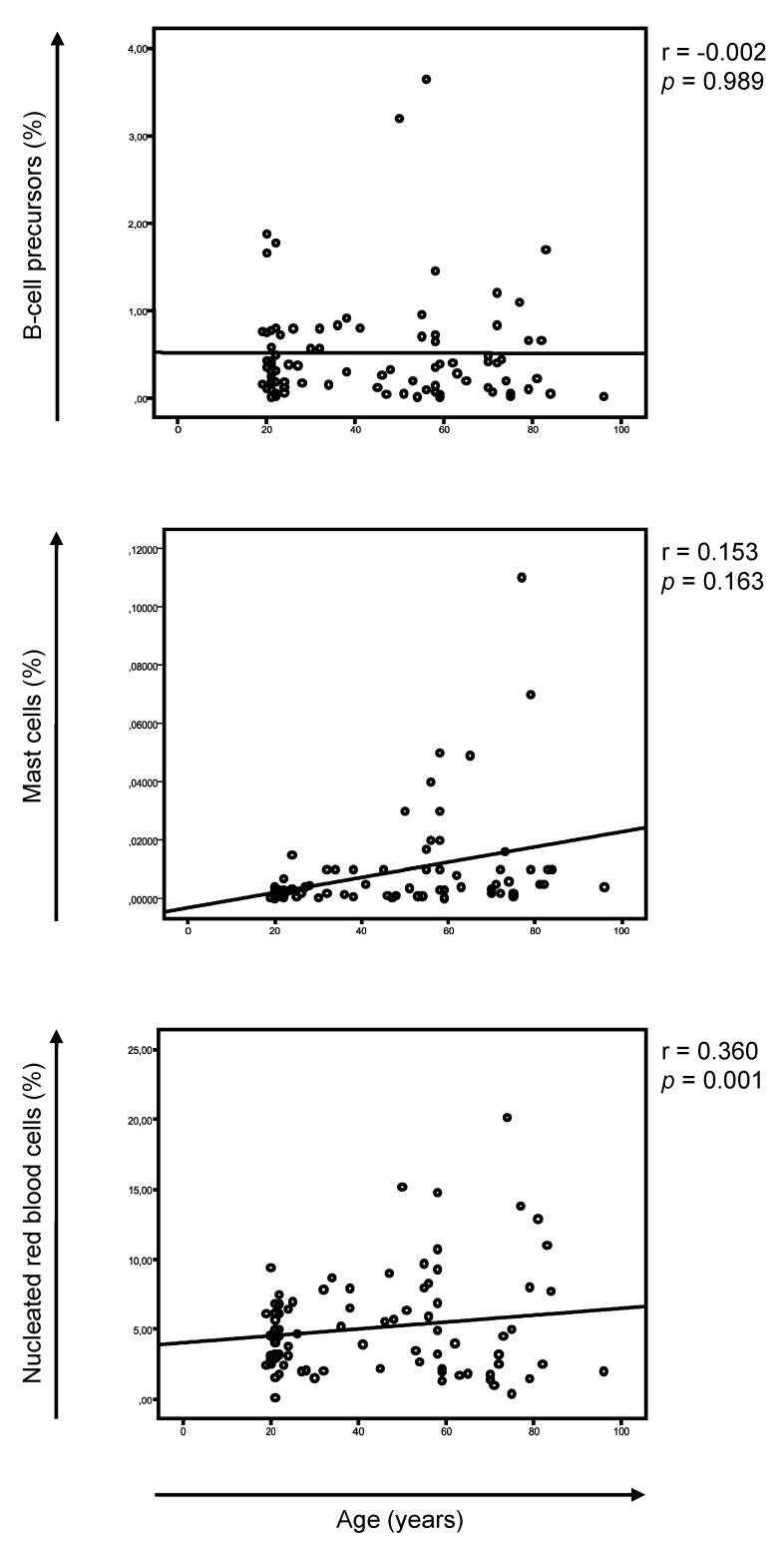
Correlation between age and the percentage of B-cell precursors, mast cells and nucleated red blood cells in bone marrow aspirates from healthy adults (*n* = 85).

**Figure 2 cancers-13-04924-f002:**
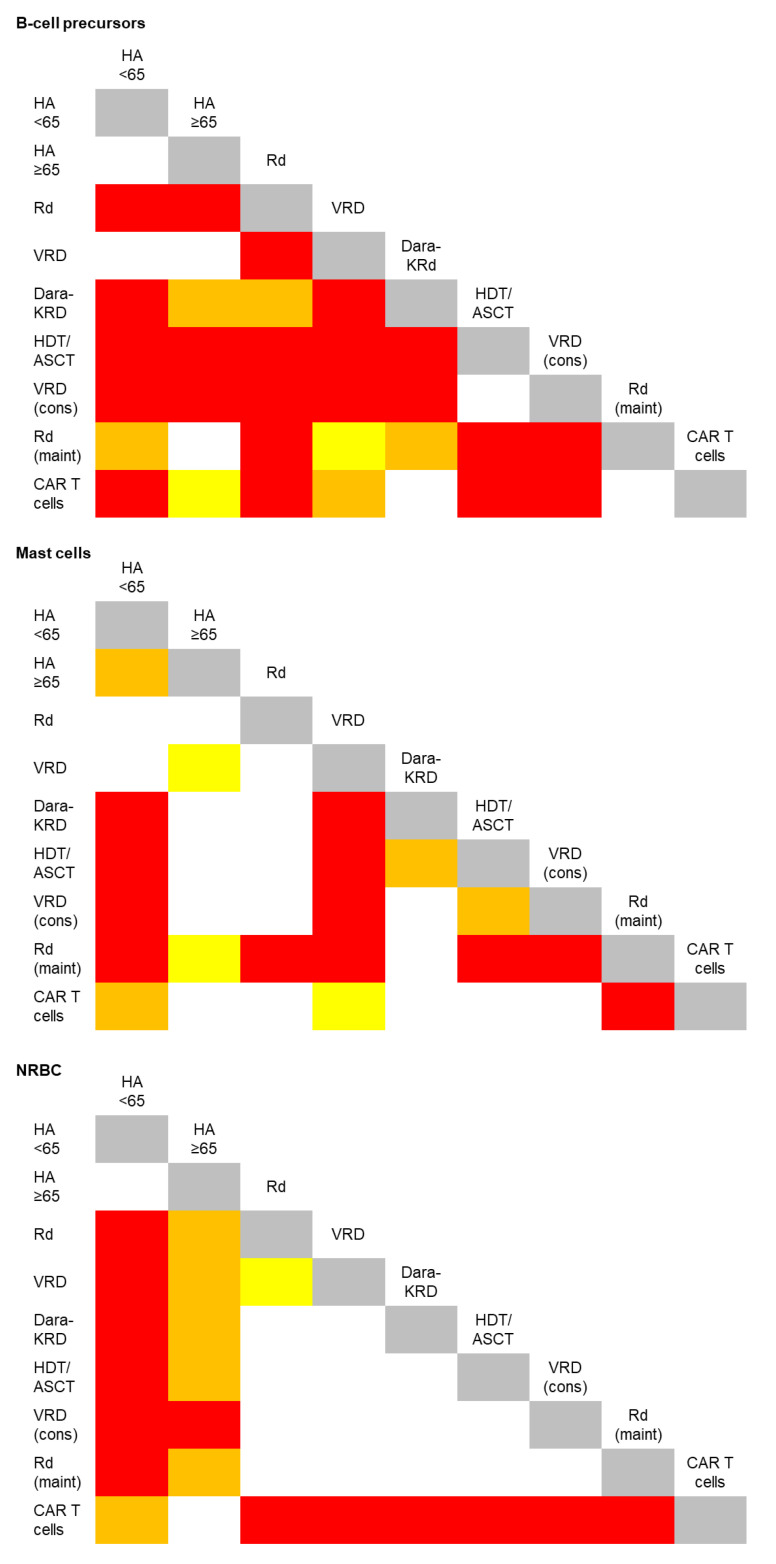
Graphical representation of significant differences in the frequency of B-cell precursors, mast cells and nucleated red blood cells (NRBC) in bone marrow aspirates from healthy adults (HA) younger and older than 65 years, as well as of patients with multiple myeloma. The latter were grouped according to treatment: induction with lenalidomide and low-dose dexamethasone (Rd) with or without clarithromycin, bortezomib, lenalidomide and dexamethasone (VRD), daratumumab, carfilzomib, lenalidomide and dexamethasone (Dara-KRd); intensification with high-dose chemotherapy followed by autologous stem cell transplantation (HDT/ASCT) and consolidation with VRD; maintenance therapy with Rd plus/minus ixazomib and salvage therapy with anti-BCMA CAR T cells. White, yellow, orange and red squares indicate *p*-values > 0.05, < 0.05, < 0.01 and < 0.001, respectively.

**Figure 3 cancers-13-04924-f003:**
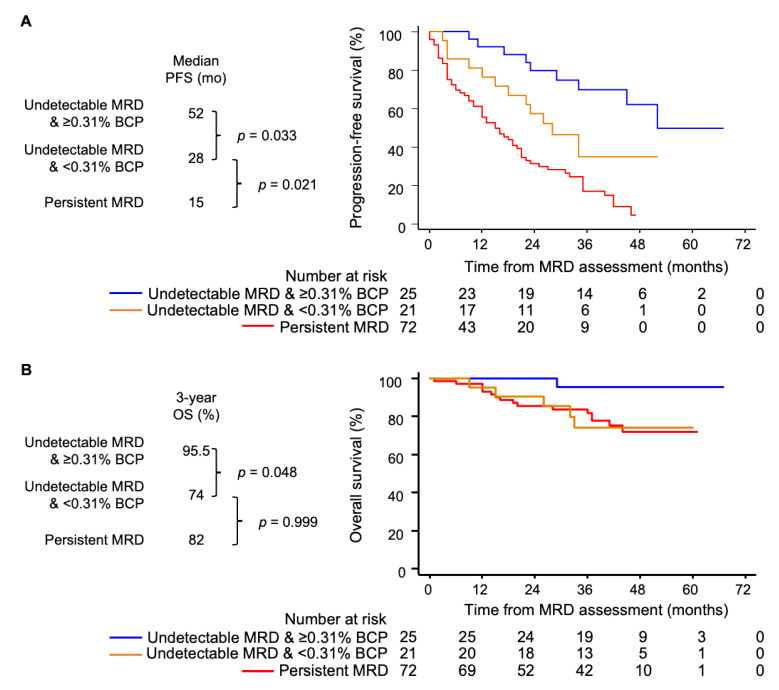
Simultaneous assessment of measurable residual disease (MRD) and hemodilution in bone marrow (BM) aspirates. (**A**,**B**) Progression-free (PFS) and overall survival (OS) of transplant-ineligible multiple myeloma patients (*n* = 118) enrolled in the PETHEMA/GEM2010MAS65 clinical trial, stratified according to detectable vs. undetectable MRD at the end of treatment and, within the latter, according to the percentage of BM B-cell precursors (BCP) below or above the median value observed in healthy adults older than 65 (0.31%, Table 1).

**Figure 4 cancers-13-04924-f004:**
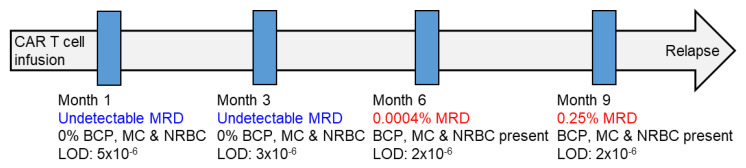
Patient case treated with anti-BCMA CAR T cells. BM aspirates were collected at months 1, 3, 6 and 9 after infusion, and the MRD assessment was performed using next-generation flow (NGF) cytometry following the EuroFlow guidelines. The first two BM aspirates were characterized by low cellularity; undetectable MRD and the absence of BCP, mast cells (MC) and nucleated red blood cells (NRNC). BM aspirates collected at months 6 and 9 overlapped with the recovery of PB cell counts and displayed increased cellularity and the presence of BCP, MC and NRBC, as well as detectable MRD. The patient progressed two months later. The limit of detection of the consecutive NGF assays is indicated in each time point.

**Table 1 cancers-13-04924-t001:** Median (range) and quartile percentages of B-cell precursors, mast cells and nucleated red blood cells in bone marrow aspirates of healthy adults grouped according to age (below or above 65 years) and patients with multiple myeloma. The latter are grouped according to treatment: induction with lenalidomide and low-dose dexamethasone (Rd) with or without clarithromycin, bortezomib, lenalidomide and dexamethasone (VRD) and daratumumab, carfilzomib, lenalidomide and dexamethasone (Dara-KRd); intensification with high-dose chemotherapy followed by autologous stem cell transplantation (HDT/ASCT) or consolidation with VRD; maintenance therapy with Rd plus/minus ixazomib and salvage therapy with anti-BCMA CAR T cells.

		Healthy Adults		Multiple Myeloma						
				Induction			Intensification		Maintenance	Salvage
		<65 (*N* = 65)	≥65 (*N* = 20)	Rd (*N* = 63)	VRD (*N* = 300)	Dara-KRd (*N* = 34)	HDT/ASCT (*N* = 364)	VRD (*N* = 350)	Rd (*N* = 232)	CAR T (*N* = 61)
B-cell precursors	Q25	0.14	0.08	0	0.03	0	0.92	0.63	0.02	0.02
	Median (range)	0.35 (0.01–3.64)	0.31 (0.02–0.66)	0.01 (0–0.90)	0.31 (0–10.0)	0.05 (0–3.1)	2.0 (0–14.5)	1.8 (0–19.6)	0.13 (0–10.7)	0.12 (0–5.75)
	Q75	0.73	0.66	0.04	1.0	0.2	3.8	4.0	0.74	0.33
Mast cells	Q25	0.001	0.003	0	0.001	0.004	0.002	0.003	0.007	0.002
	Median (range)	0.002 (0.0002–0.05)	0.005 (0.001–0.11)	0.0095 (0–0.14)	0.004 (0–0.93)	0.01 (0–0.4)	0.006 (0–0.11)	0.01 (0–0.26)	0.01 (0–0.76)	0.007 (0–0.17)
	Q75	0.007	0.01	0.03	0.01	0.04	0.01	0.02	0.06	0.01
Nucleated red blood cells	Q25	2.7	1.7	4.0	4.4	6.0	4.5	5.0	3.5	1.8
	Median (range)	4.6 (0.1–15.2)	2.50 (0.4–20.1)	11.0 (0–39.0)	7.3 (0–43.0)	7.3 (1.8–26.4)	8.2 (0–53.9)	8.2 (0–53.9)	7.5 (0–52)	2.5 (0–28.4)
	Q75	6.8	7.9	19.3	12.6	10.8	13.0	13.8	12.6	4.5

**Table 2 cancers-13-04924-t002:** Median (range) and quartile percentages of B-cell precursors, mast cells and nucleated red blood cells in bone marrow aspirates of patients with multiple myeloma enrolled in PETHEMA/GEM2010MENOS65. Patients were grouped according to their MRD status at the end of the treatment.

		*N* = 118	MRD Status	
			Positive (*N* = 72)	Negative (*N* = 46)
B-cell precursors	Q25	0.13	0.18	0.09
	Median (range)	0.42 (0–7.7)	0.48 (0–7.7)	0.33 (0–4.6)
	Q75	1.46	1.36	1.51
Mast cells	Q25	0.01	0.01	0.005
	Median (range)	0.02 (0–0.55)	0.02 (0–5.5)	0.015 (0–0.33)
	Q75	0.07	0.07	0.07
Nucleated red blood cells	Q25	9.56	10.7	8.74
	Median (range)	17.4 (0–50.9)	16.0 (1.4–50.9)	17.6 (0–45)
	Q75	26.7	25.2	28.1

**Table 3 cancers-13-04924-t003:** Frequency of bone marrow aspirates with undetectable B-cell precursors, mast cells and nucleated red blood cells (NRBC). Data is shown per cell type, as well as the absence of any of the three, two or all three cell types. Healthy adults (HA) were grouped according to age (younger and older than 65), whereas patients with multiple myeloma were grouped according to treatment: induction with lenalidomide and low-dose dexamethasone (Rd) with or without clarithromycin, bortezomib, lenalidomide and dexamethasone (VRD), daratumumab, carfilzomib, lenalidomide and dexamethasone (Dara-KRd); intensification with high-dose chemotherapy followed by autologous stem cell transplantation (HDT/ASCT) and consolidation with VRD; maintenance therapy with Rd plus/minus ixazomib and salvage therapy with anti-BCMA CAR T cells.

	Healthy Adults		Multiple Myeloma							*p*-Value
			Induction			Intensification		Maintenance	Salvage	
	<65 (*N* = 65)	≥65 (*N* = 20)	Rd (*N* = 63)	VRD (*N* = 300)	Dara-KRd (*N* = 34)	HDT/ASCT (*N* = 364)	VRD (*N* = 350)	Rd (*N* = 232)	CAR T (*N* = 61)	
BCP	0 (0%)	0 (0%)	27 (45%)	32 (11%)	9 (26.5%)	11 (3%)	10 (3%)	31 (14%)	10 (17%)	<0.001
MC	0 (0%)	0 (0%)	16 (27%)	48 (16%)	1 (3%)	32 (9%)	27 (8%)	28 (12%)	9 (15.5%)	<0.001
NRBC	0 (0%)	0 (0%)	3 (5%)	2 (0.7%)	0 (0%)	8 (2%)	6 (2%)	5 (2%)	7 (12%)	<0.001
One of three	0 (0%)	0 (0%)	24 (12%)	59 (20%)	8 (23.5%)	31 (8.5%)	24 (7%)	42 (19%)	8 (14%)	<0.001
Two of three	0 (0%)	0 (0%)	11 (18%)	10 (3%)	1 (3%)	1 (0.3%)	2 (0.6%)	8 (4%)	3 (5%)	
All three	0 (0%)	0 (0%)	0 (0%)	1 (0.3%)	0 (0%)	6 (2%)	5 (1.4%)	2 (1%)	4 (7%)	

## Data Availability

For the original data, please contact noemipuig@usal.es.

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
