# Peer review of "Reference Values to Assess Hemodilution and Warn of Potential False-Negative Minimal Residual Disease Results in Myeloma"

_cancers, 2021, doi:10.3390/cancers13194924_

Round 1

Reviewer 1 Report

The manuscript is about increasing the value of MRD measurements by using bone marrow cell populations to inform on hemodilution. The topic is novel, the approach is very interesting, and the manuscript is in generally well written. I congratulate the authors with the study and think this is worthy of publication. There are however some issues

One thing I miss is that the authors state that hemodilution should preferably be evaluated with references obtained from patients treated with identical regimens. But then, the only evaluation of PFS/OS comparing cell populations over or under a reference, is performed against healthy adults. If it is preferable to use the reference from identical regimens, why wasn’t this performed? I also wonder why PFS/OS data were only presented for one study, of the several studies part of this study. Were the PFS/OS data not in line with the hypothesis for the other studies?

In the introduction (line 63-66) it is speculated in what causes MRD-negative patients to progress fast. An obvious omission here is that there could be patients in “real” MRD-negativity, where the aggressiveness and high proliferation of the disease leads to a swift relapse. Should be mentioned. It’s not all necessarily about sensitivity/representativeness

I miss some discussion topics.

  • What to do with the patients that are MRD negative but with suspected hemodilution? They could be either real MRD-negative, or false MRD-negative. Mass-spec could be mentioned as an upcoming method circumventing this problem. Or should there be a new sample performed in a different location? Are there suggestions of additional evaluations to confirm/refute that there is hemodilution?
  • Is it possible that lack of precursors is in many patients not because of hemodilution, but because of “empty”/suppressed/non-functioning bone-marrow, which indirectly can be a cause of early progression, either because of lack of immune effectors or because of less tolerance for continuous treatment? With a missing gold-standard for hemodilution, this remains a question, which is not discussed.

Regarding the CART-cell patient(s):

Line 230-232: That CAR-T cells-treated marrows are lymphodepleted are not the same as dry BM aspirates. I’m not aware of data showing that there are more dry aspirates in CART-treated patients than in others, and it is not referenced here either. I don’t see any documentation that aspirates from these patients are more hemodiluted than other patients’. Without any such documentation or reference, I think this is too speculative.

Line 232-235 and figure 3: I think this is not in line with just stating that CART-marrows often have absence of all three cell types. And I don’t think this particular case is very informative. This could just as well be representative bone marrow also in the first two samples, with depleted marrow, and real MRD negativity. And then the patient progresses along with the normalization of the marrow, without any impact of hemodilution. So I don’t think this is a proof-of-principle example.

Small points:

Line 26-27: First sentence does not read well, at least change “negative” to “negativity”, but should probably be rephrased. Start of the abstract (line 37) is better.

Line 56 and beyond: I thought that references in brackets ([1-6]) should be before the period/dot, and not after.

“Materials and methods”. This section contains several sub-headings, for MRD assessment, NGF and statistics. But not for the first half of the section. Should either be for all topics, or for none.  

Line 216: “55months not reached”. Does this mean that after 55m median follow-up the median PFS has not been reached? Normally this is just stated as non-reached. So a little bit confusing, but the information is interesting. Is there another way of phrasing it? For instance “28 vs non-reached, after 55months of follow-up”

Author Response

Reviewer 1

Reviewer 1 general comment: The manuscript is about increasing the value of MRD measurements by using bone marrow cell populations to inform on hemodilution. The topic is novel, the approach is very interesting, and the manuscript is in generally well written. I congratulate the authors with the study and think this is worthy of publication. There are however some issues.

Answer to the Reviewer 1 general comment: We are thankful for the positive opinion about our manuscript. Also by the comments made by the Reviewer, which were addressed below and substantially improved its quality.

Reviewer 1 specific comment 1:  One thing I miss is that the authors state that hemodilution should preferably be evaluated with references obtained from patients treated with identical regimens. But then, the only evaluation of PFS/OS comparing cell populations over or under a reference, is performed against healthy adults. If it is preferable to use the reference from identical regimens, why wasn’t this performed? I also wonder why PFS/OS data were only presented for one study, of the several studies part of this study. Were the PFS/OS data not in line with the hypothesis for the other studies?

Answer to the Reviewer 1 specific comment 1:  Please note that the evaluation of PFS/OS comparing cell populations over or under a reference of healthy adults, was deliberately performed to evaluate the applicability of hemodilution assessment in treatment scenarios where a reference based on patients receiving the same regimen was unavailable. Also because of the trial design, since MRD and hemodilution were assessed at the end of induction therapy and there was no further treatment thereafter, which is an obvious confounding variable. It should be noted that the majority of clinical trials used to define reference values include treatment until disease progression and still have relatively short follow up. Nevertheless and because of the Reviewer’s comment, we applied the same strategy in patients assessed after consolidation (and before maintenance) in the GEM2012MENOS65 clinical trial. We found no significant differences in survival outcomes between MRD negative patients if stratified based on percentages below or above the respective median value of B cell precursors, mast cells and NRBC found in the same population. These results have been added in the revised version of the manuscript together with the statement that the utility of the reference values specific of certain regimens provided in the manuscript should be confirmed by other groups (lines 317-326).

Reviewer 1 specific comment 2:  In the introduction (line 63-66) it is speculated in what causes MRD-negative patients to progress fast. An obvious omission here is that there could be patients in “real” MRD-negativity, where the aggressiveness and high proliferation of the disease leads to a swift relapse. Should be mentioned. It’s not all necessarily about sensitivity/representativeness.

Answer to the Reviewer 1 specific comment 2:  The Reviewer is correct and the fact that the biology of the disease may lead to a swift relapse was added in the revised version of the manuscript (lines 68-69).

Reviewer 1 discussion topic 1:  What to do with the patients that are MRD negative but with suspected hemodilution? They could be either real MRD-negative, or false MRD-negative. Mass-spec could be mentioned as an upcoming method circumventing this problem. Or should there be a new sample performed in a different location? Are there suggestions of additional evaluations to confirm/refute that there is hemodilution?

Answer to the Reviewer 1 discussion topic 1:  The Reviewer is again correct that in patients with suspected hemodilution an MRD negative result can be either real or false. Because of the Reviewer’s suggestion, the possibility of performing mass-spec before repeating a bone marrow aspirate and the need of a new bone marrow sample if the M-protein remains undetectable using more sensitive methods was added in the Conclusion section of the revised version of the manuscript (lines 426-429).

Reviewer 1 discussion topic 2: Is it possible that lack of precursors is in many patients not because of hemodilution, but because of “empty”/suppressed/non-functioning bone-marrow, which indirectly can be a cause of early progression, either because of lack of immune effectors or because of less tolerance for continuous treatment? With a missing gold-standard for hemodilution, this remains a question, which is not discussed.

Answer to the Reviewer 1 discussion topic 2: Indeed, lack of precursors could result either because of hemodilution or because of disrupted hematopoiesis. Because of the Reviewer’s comment, we have added in the Conclusion section of the revised version of the manuscript that the lack of a given cell type can result by either phenomenon, and that in the absence of a gold-standard for hemodilution, greater specificity in determining hemodiluted samples could potentially be achieved by considering only those in which plasma cells, B cell precursors, mast cells and NRBC are simultaneously absent (lines 421-425).

Reviewer 1 discussion topic regarding the CART-cell patient(s):

Line 230-232: That CAR-T cells-treated marrows are lymphodepleted are not the same as dry BM aspirates. I’m not aware of data showing that there are more dry aspirates in CART-treated patients than in others, and it is not referenced here either. I don’t see any documentation that aspirates from these patients are more hemodiluted than other patients’. Without any such documentation or reference, I think this is too speculative.

Line 232-235 and figure 3: I think this is not in line with just stating that CART-marrows often have absence of all three cell types. And I don’t think this particular case is very informative. This could just as well be representative bone marrow also in the first two samples, with depleted marrow, and real MRD negativity. And then the patient progresses along with the normalization of the marrow, without any impact of hemodilution. So I don’t think this is a proof-of-principle example.

Answer to the Reviewer 1 discussion topic regarding the CART-cell patient(s): Please note that rather than mere depletion of lymphocytes, we usually observe disappearance of virtually all bone marrow cell types in the early stages after CAR T cell infusion. Hence, our comment that these samples resembled dry bone marrow aspirates. We agree with the Reviewer that there is no documentation on this and that it is an observation based on the routine assessment we perform in these patients; accordingly, the mention of dry BM aspirates was deleted in the revised version of the manuscript.

            The patient case represented in the figure is very illustrative of what we commonly observe in the early stages after CAR T cell infusion. We believe that regardless of the reason why these cells are absent, it does illustrate that these samples are not empowered for high sensitive MRD assessment. Therefore, we believe that albeit being preliminary, a patient case may alert other investigators about the limitations of MRD testing soon after CAR T cell infusion. Please note that larger studies in patients enrolled in pivotal clinical trials are being conducted, and will confirm that the most relevant time points for MRD assessment are near to 12 months after CAR T cell infusion.

Reviewer 1 small points:

Line 26-27: First sentence does not read well, at least change “negative” to “negativity”, but should probably be rephrased. Start of the abstract (line 37) is better.

Line 56 and beyond: I thought that references in brackets ([1-6]) should be before the period/dot, and not after.

“Materials and methods”. This section contains several sub-headings, for MRD assessment, NGF and statistics. But not for the first half of the section. Should either be for all topics, or for none. 

Line 216: “55months not reached”. Does this mean that after 55m median follow-up the median PFS has not been reached? Normally this is just stated as non-reached. So a little bit confusing, but the information is interesting. Is there another way of phrasing it? For instance “28 vs non-reached, after 55months of follow-up”

Answer to the Reviewer 1 small points: Thank you for alerting us to and all the points raised by the Reviewer have been corrected in the revised version of the manuscript. Please note that a “vs” was missing in the sentence about patients’ PFS. It now reads as: Patients with undetectable MRD and <0.31% B-cell precursors displayed significantly inferior PFS vs MRD negative cases with ≥0.31% B cell precursors (median of 28 vs 55 months vs not reached, P = .033; Figure 3A)

Reviewer 2 Report

In the paper entitled: “Reference Values to Assess Hemodilution and Warn of Potential False-Negative Minimal Residual Disease Results in Myeloma” Noemi Puig and al. reiterate the need to evaluate the presence of any hemodilution in bone marrow aspirate samples obtained from subjects with MM intended for evaluation of MMR in order to avoid false negatives.

The authors supports the routine assessment of BM cellularity to assess hemodilution while evaluating MRD.

The work is very interesting, well conducted and the evaluation is performed in various MM settings and compared with bone marrow aspirates obtained from healthy subjects. Major revisionThe authors  furnish provide detailed information about their analysis and results however  they should also provide more objective elements enriching the manuscript with representative figures, cytograms and dot plots that may better  clarify methodologically how to proceed with the evaluation of the parameters used for   quantifying the  hemodiluition.In particular, in consideration of the importance of adequately assessing the  haemodilution level in samples intended for MRD studies, the authors should provide more information about the analysis of the parameters used so that the methodology used can also be reproduced in other laboratories.

Author Response

Reviewer 2

Reviewer 2 general comment: In the paper entitled: “Reference Values to Assess Hemodilution and Warn of Potential False-Negative Minimal Residual Disease Results in Myeloma” Noemi Puig and al. reiterate the need to evaluate the presence of any hemodilution in bone marrow aspirate samples obtained from subjects with MM intended for evaluation of MMR in order to avoid false negatives. The authors supports the routine assessment of BM cellularity to assess hemodilution while evaluating MRD. The work is very interesting, well conducted and the evaluation is performed in various MM settings and compared with bone marrow aspirates obtained from healthy subjects.

Answer to the Reviewer 2 general comment: We are thankful for the positive opinion about our manuscript. Also by the major revision suggested by the Reviewer, which was addressed and substantially improved the quality of the revised version of the manuscript.

Reviewer 2 major revision: The authors  furnish provide detailed information about their analysis and results however  they should also provide more objective elements enriching the manuscript with representative figures, cytograms and dot plots that may better  clarify methodologically how to proceed with the evaluation of the parameters used for   quantifying the  hemodiluition. In particular, in consideration of the importance of adequately assessing the  haemodilution level in samples intended for MRD studies, the authors should provide more information about the analysis of the parameters used so that the methodology used can also be reproduced in other laboratories.

Answer to the Reviewer 2 major revision: Because of the Reviewer’s comment, we have added dot plots indicating how to identify the cell types indicative of hemodilution in the new Supplemental Figure 1, as well as information on the markers that should be used for the identification of such cell types in the Conclusion section of the revised version of the manuscript (lines 415-416). Please note that their phenotype was already described in the Methods section of the original version of the manuscript

This manuscript is a resubmission of an earlier submission. The following is a list of the peer review reports and author responses from that submission.